# Remarks on Geomatics Measurement Methods Focused on Forestry Inventory

**DOI:** 10.3390/s23177376

**Published:** 2023-08-24

**Authors:** Karel Pavelka, Eva Matoušková, Karel Pavelka

**Affiliations:** Department of Geomatics, Faculty of Civil Engineering, Czech Technical University in Prague, 166 29 Prague, Czech Republic; eva.matouskova@fsv.cvut.cz (E.M.);

**Keywords:** UAV, RPAS, photogrammetry, laser scanning, mobile laser scanning, lidar sensor, ALS, PLS, TLS, forestry, DBH

## Abstract

This contribution focuses on a comparison of modern geomatics technologies for the derivation of growth parameters in forest management. The present text summarizes the results of our measurements over the last five years. As a case project, a mountain spruce forest with planned forest logging was selected. In this locality, terrestrial laser scanning (TLS) and terrestrial and drone close-range photogrammetry were experimentally used, as was the use of PLS mobile technology (personal laser scanning) and ALS (aerial laser scanning). Results from the data joining, usability, and economics of all technologies for forest management and ecology were discussed. ALS is expensive for small areas and the results were not suitable for a detailed parameter derivation. The RPAS (remotely piloted aircraft systems, known as “drones”) method of data acquisition combines the benefits of close-range and aerial photogrammetry. If the approximate height and number of the trees are known, one can approximately calculate the extracted cubage of wood mass before forest logging. The use of conventional terrestrial close-range photogrammetry and TLS proved to be inappropriate and practically unusable in our case, and also in standard forestry practice after consultation with forestry workers. On the other hand, the use of PLS is very simple and allows you to quickly define ordered parameters and further calculate, for example, the cubic volume of wood stockpiles. The results from our research into forestry show that drones can be used to estimate quantities (wood cubature) and inspect the health status of spruce forests, However, PLS seems, nowadays, to be the best solution in forest management for deriving forest parameters. Our results are mainly oriented to practice and in no way diminish the general research in this area.

## 1. Introduction

Dendrometry deals with the determination of the quantitative parameters of standing trees and forest stands, the relationship between these parameters, and the development of methods for capturing them. The main task of dendrometry is to quantify the volumes of individual trees and stockpiles of whole stands, to measure intermediate quantities, and to develop and use procedures and methods for determining the volumes and stocks of wood. Forestry and agriculture are traditional users of photogrammetry and remote sensing. In the last decade, other technologies have been used, such as terrestrial laser scanning (TLS), aerial laser scanning (ALS), and mobile laser scanning (MLS). These technologies can supply map data, with remote sensing providing further qualitative information on vegetation. Quantitative forestry information (cubic capacity of wood) has so far been obtained using ground analyses and calculations using evaluation tables. The article discusses methods for using possible sensors in forest inventory. The methods analyzed are classical methods, laser scanning with different devices, aerial methods, photogrammetry, the use of drones, SLAM technology (simultaneous localization and mapping), and mobile mapping methods, as well as cheap methods using smartphones and tablets. All this is performed on the example of a selected part of a spruce forest in the mountains area in the Czech–German border region.

### 1.1. Traditional Approach

The selected parameters of forest cover, which are used for calculating or extrapolating wood mass quantities (cubic capacity or volume), can be obtained traditionally via measurements by hand during terrain visits or in a non-contact way (indirectly through other measurements and quantities).

The basic parameters for the computation of wood cubature are as follows:(a)The diameter at breast height (DBH), measured using a caliper in two rectangular directions by hand;(b)Approximate tree height using measuring instruments based on simple measurement devices based on triangulation;(c)Density of growth.

These parameters are common and traditional in forestry practice, and they have been used for a long time. According to [1], the accuracy of the wood volume determination when using a caliper and average tree height for all trees is fully ±5%.

The heights are measured on a smaller set of selected trees only, and ultimately, the total stock is determined by volume tables, volume curves, or the sample methods.

### 1.2. Modern Technology Implementing

In the last decade, forestry has been trying to adapt to new trends in modern technology. The most important role in forest taxation is the measurement of basic dendrometric quantities. New devices are rapidly being integrated, and new methods are being explored that could bring about the simplification and streamlining of certain procedures of work [2].

Conventional ground measurement methods are still a priority in practice; introducing new optical, digital, and laser devices can increase their accuracy, but also the price of measurement. Still, in the last few years, new methods have been increasingly experimental, with new innovations for field measurements.

Practical, non-traditional methods of measurement can be accurate, but they are usually very expensive in terms of device price, and importantly, they are time consuming with regard to data processing. After all, data processing is not trivial and requires experts. From the point of view of time saving and the minimization of personnel, non-contact or general remote sensing methods are still in development, and new technologies and instruments are being examined [3,4]. An overview of the existing methods is given below. Crucial for the use of new technologies in practice are their price, data processing requirements, and especially the size of the assessed area. The administrators of small private forest areas will have completely different requirements compared to large owners or state-owned areas.

### 1.3. Photogrammetry

Apart from classical measuring methods, the other oldest data collection technology used is certainly photogrammetry. Numerous recent publications have dealt with photogrammetric methods used in agriculture and forestry. Most publications are devoted to aerial or drone surveying of the quality of agricultural land, the state of irrigation, the health status of vegetation, and the presence of pests [5,6]. The situation is similar in forestry; the major difference is the height and shelf life of forest stands, unlike agricultural crops. From this point of view, photogrammetry is a traditional, non-contact technology for spatial data collection. In forestry, it is used mainly in the form of aerial photogrammetry, especially for monitoring the conditions of forest stands and storm damage and the control of afforestation and logging. Classical analogue terrestrial photogrammetry is not used in forestry. After switching to digital photogrammetry and after the rapid development of computer technology at the beginning of the new century, photogrammetric automated IBMR technology has expanded significantly. This has been through the development of software for 3D information extraction from photos (nowadays, the most well-known software is Agisoft Photoscan/Metashape 1.7.4, built 13028 2021 (64bit), but there are many other kinds of software for automated close-range digital photogrammetry). The automatic processing of photographic images into 3D models is still very popular in general, and specialized projects can be found in forestry too [7]. 

A big and laborious international project focused on using IBMR was carried out in the Slovak Republic and Austria [8]. The typical use of IBMR photogrammetry can be found in RPAS photogrammetry. Basic RPAS’s are equipped with a simple RGB (red-green-blue) camera, which is used for orthophoto production and a DSM (digital surface model), or a textured mesh model [9,10,11].

### 1.4. Satellite, Aerial, and RPAS Remote Sensing

For over 50 years, satellite remote sensing has been successfully used in many branches [12]. The price of data is decreasing, and moreover, their spectral range, geometrical, and temporal resolution are increasing. Satellite remote sensing was used in forestry and agriculture long ago, practically from the beginning of its civilian use. It has been said that satellite technology is mainly suitable for large areas. This also applies to forestry. However, a new type of submeter satellite data can be purchased for relatively small areas (25 km^2^) too. Of course, a small owner who owns several hectares of forest will probably not use satellite data. Aerial remote sensing is generally more expensive than satellite remote sensing, but it gives a better resolution and can be more operational; often, by using specialized aircraft, different instruments such as hyperspectral scanners, laser scanners, and other devices are used. Relatively new is low-cost ultralight aircraft or RPAS (drone) remote sensing [10,11,13]. Ultralights can be equipped with several professional instruments, but their operating range is bounded, as is the flight range of drones. Professional drones are nowadays equipped with multispectral, thermal, infrared, and RGB cameras, and in some cases, with miniature hyperspectral or laser scanners too [14,15,16,17]. The detection of individual trees is relatively easy, especially in orchards or plantations [18].

### 1.5. Laser Scanning

At the beginning of the new millennium, TLS appeared [19]. Much research and many projects have been focused on laser scanning in general; this also applies to forestry [20,21,22,23,24]. However, using TLS in forestry is relatively new; projects related to using TLS for determining a typical tree (for modeling), or for measuring DBH, have been carried out mainly in the last decade [25,26,27,28]. It is necessary to say that there are two points of view on the matter: the scientific one and the practical one. Not all scientific results can be used in forestry practice.

### 1.6. Aerial Laser Scanning and Mobile Laser Scanning

In the late 1990s, ALS started being used; this technology proved to be effective in creating DSMs and DRMs (digital relief model), and it is used quite regularly [29].

ALS is a very good technology for creating DSMs; in forestry, this means the digital representation of forest, that is, the forest surface model. For example, the canopy height from the lidar data was examined in [30]. Tropical rainforest structures using ALS were analyzed in [31]. The delineation of a forest using ALS data was tested in [32]. From at least two overflights (and two DSMs) made at different times (typically in the span of some years), it is possible to obtain the volume increases in wood mass [33]. ALS gives us a better or worse DSM only, the quality of which depends on the instrument used and the type of forest. From the DSM, it is possible to calculate the average height of the forest and its area. If we know the average number of trees per unit area and the approximate DBH, we can calculate the wood volume. However, the density of the point cloud derived from ALS is crucial. Typically, for creating a DRM used in topography (for example, in the Czech Republic by the government), the point density reaches 1–3 points on m^2^, which is too low for forest DSM modeling. Special overflights with much a better point density are expensive. Low-cost ALS technology can be the solution. However, from ALS, we obtain data from above and not from inside the analyzed forest. 

MLSs, originally mounted on the car roofs of road vehicles, have been used since the beginning of the new century. Their price is very high and the use of them in forested terrain is not possible. For example, a Riegl laser scanner with a GNSS (global navigation satellite system) held on a tractor has been used for tree diameter estimation [34]. With the expansion and price reduction in the inertial measurement unit (IMU), MLS systems have changed to smaller, cheaper, handheld, or indoor systems.

Simpler MLS systems seem to be a good solution for forest inventory and forest resource management. However, there are problems with forest possibility assessments, even for small, motorized vehicles (e.g., ATVs) [35]. 

Handheld or backpack instruments [36] are better, of course, only for small areas. PLSs have been intensively used recently [35,37,38,39,40,41]. These systems are still in development, but they have a future.

### 1.7. Mobile Low-Cost Systems (Smartphones and Tablets)

One new method is to implement measurement devices into common instruments such as smartphones or tablets. Using a mobile phone with RGB-D SLAM for obtaining basic parameters such as DBH, the height of trees, and tree position was analyzed [42]. 

In 2020, Apple launched the new iPad Pro, with a small laser scanner. Maybe in the future, these instruments will be suitable for measurements in forestry due to them being inexpensive and easy to use compared to equipment that is expensive, often heavy, and difficult to process. Lidar on an iPhone or tablet today uses the viDOC Pix4D system to create high-quality 3D models of smaller objects using the GNSS RTK module [43,44].

### 1.8. Summary of the State of Knowledge

In the last twenty years, technologies have changed dramatically. Many authors have attempted to use terrestrial laser scanning in all forms and IBMR photogrammetry in forestry, with varying degrees of success and possibilities for implementation in practice [7,8,9,11,20]. Other methods such as ALS or satellite remote sensing have been used too [33]. Multispectral or hyperspectral satellite or aerial remote sensing is used to determine the type of forest and its condition based on the specific reflectivity of electromagnetic radiation, but that is a different kind of information. ALS can also be included in remote sensing methods.

ALS is successfully used for DSM and DRM acquisition in many countries, mainly for mapping. For example, the whole area of the Czech Republic was measured using ALS from 2009 to 2013 with a point density of 1–2 pts/m^2^, which is perfect for cartography and creating a DRM, but not good enough for other work such as archaeology, city modeling, and tree detection in forestry, etc. In forestry, the point density in ALS technology should be orders of magnitude larger, and it is not possible to be used for whole-state territories. This means that special projects must be carried out for the above-mentioned branches, and, of course, for relatively small areas (tens to hundreds of square kilometers), which is quite expensive. 

Aerial stereophotogrammetry has been used for a long time as a basic mapping technology. After the digitization of aerial photogrammetry, computing a DSM became possible, but its massive use is relatively new because of hardware performance and software. A classical DSM from a forested area is typically very noisy and can be used mainly for detecting the approximate height of the measured vegetation. Some research is focused on calculating trees, but it is often on a cultivated plantation of trees (e.g., palm trees), which are planted regularly, and the palm trees are very similar [18]. Calculating tree numbers from aerial images or an orthophoto in a common mixed forest is difficult due to the diversity of species and involvement of tree crowns of a particularly deciduous forest. Calculating trees works better in coniferous forests. Today’s RPAS capabilities can solve this problem in some cases.

### 1.9. Project Aims

The aim of this study was to find the optimal procedure for obtaining a 3D model inside a forest to derive the forest stand parameters. It meant to find and analyze suitable new methods for forestry measurements using the modern available top technologies in terms of their price, time, precision, and usability in practice. In this case, mainly digital automated photogrammetry IBMR, TLS with different scanners, low-cost ALS based on small aircraft, MLS, and RPAS (drones) were used. 

## 2. Study Area

Manual (traditional) measurements are still in use as a basic technology. Based on our preliminary research, for the time being, we excluded the measurement for a deciduous forest. As a test area, a small part of a commonly grown spruce forest (*Picea abies* [L.] *Karst*) in the Ore Mountains (near the Czech–German border in the northwest of the Czech Republic) was selected (50.37 N, 12.86 E). It is possible to say that, in this case project; a typical Czech coniferous forest was used. 

The elevation of the site was 930 m, with the size of the test area being approximately 35 × 30 m. A part of this site was a mountain meadow, which allowed us to measure some parameters such as tree height using geodetic instruments (total station Trimble 5000 Series); the approximate height of the trees was between 20 m and 25 m, and the diameter (DBH) was from 20 cm to 50 cm.

Seventy-six trees in total were marked and measured in a time span of about three hours. For a better orientation, the tested area was marked by plastic tape, and the measured trees were marked with chalk. The selected area was measured and analyzed using traditional forestry practice, meaning that the measurements were performed with a caliper, etc. These results were obtained for a comparison with other methods and were used as reference data [1,2].

First, the average diameter of all the trees (76 trees altogether) in the study area (DBH) was measured using a caliper. The average measured thickness of the stems was 29.4 cm. The second measurement was only for eight typical stems; thus, the average from the eight measurements was 30.4 cm. Third, the height of the typical trees was measured geodetically; the average height was 22.60 m. 

## 3. Methodology—Derivation of Dendrometry Parameters

Experiments were performed to measure the DBH, the number of trees, and their height and wood volume in a section of forest. 

The study area was measured with all possible technologies—terrestrial close-range photogrammetry (IBMR), TLS, RPAS (drone) photogrammetry, MLS/PLS, ALS, and with a tablet equipped with a lidar sensor.

### 3.1. IBMR Photogrammetry

In general, terrestrial photogrammetry has often been tested to determine the thickness, shape, or volume of tree stems. Nowadays, a very popular and frequently used technology in many fields for creating a 3D model from photographs, generally called IBMR, uses various techniques to solve problems, such as SfM (structure from motion) and MVS (multi view stereo). Current computer IBMR technology can process a large volume of photos and generate point clouds automatically. The procedure of finding the thickness of tree stems by processing photographic images is presented in [7,8], and when the RMSE (root-mean-square error) of the calculated thicknesses of the stems was compared to the measured thickness, the result was 2.4 cm. 

Based on the literature research, IBMR photogrammetry was applied to our study area as the simplest and cheapest method. 

#### On-Site Tests

Together, six tests were carried out on the same test area (Figure 1) with different cameras and different imaging parameters over two years during 2018–2020 (Table 1). 

During test No.1, photographs were taken in rows, with the axes of the photos being perpendicular to the rows. The step between the photographs was more than 1 m, which seemed too big; a step of only 20 cm was used in [8]. Big problems were wind, low lighting in cloudy weather, shadows in sunny weather, and the forest condition (tidiness), which is not usually mentioned in the literature. Due to the movement of tree crowns, it was not possible to create a complex 3D model (this problem occurs using photogrammetry, laser scanning, and RPAS too). In some parts of our forested study area, it was not possible to take photos in ideal lines or distances between the photo positions. From 280 photos, less than 5% were processed and the model was not sufficient.

Thus, the assumption was that the model quality depended on the tree density, terrain structure, the direction of the photo axis, and mainly on the number of photos and depth of the sharpness (which was very important in this case).

During the second test, a different camera with a wide lens was used, and from the 244 photos that were taken, only 10% were successfully processed. In both experiments, the image overlapping was not good and the distances between the photo positions were too big.

After both unsuccessful attempts, other experiments were carried out. A HP workstation Z240 (32 GB RAM, SSD HDD, i7 processor 7700, 3.6 GHz, graphic card Nvidia Quadro P2000 with Metashape acceleration) was used for new complex data processing. 

The experiments were prolonged with different parameters with the aim of obtaining a sufficient 3D model. During test No.3, the distances between the photo stations were from approximately 50 cm to 1 m. In total, 646 of 689 photos were oriented (processing time 5 h). The computed dense cloud had more than 200 million points. It can be said that, upon first view, this experiment was successful. However, after the dense point processing, a mesh could not be processed due to memory problems (too many points); therefore, this is not applicable to normal practice (Figure 2 and Figure 3).

The fourth experiment used a camera with a different lens, because the wide lens proved to be inappropriate for this type of measurement. However, only 131 images of 775 were oriented to a dense point cloud containing 67 million points. Nine plastic control points were added into the forest, but they did not improve the processing (Figure 4). 

The fifth experiment used a different type of compact camera with a GNSS module. Despite a large number of photos (more than 500) and a short photo basis (from 40 cm to 80 cm), the process failed to calculate such a quality model.

The last, sixth test was built on the relative success of the third test. Unfortunately, only 368 images of 631 were oriented, and the dense point cloud had 81 million points. Some subsets were oriented separately (two chunks with 148 and 80 images, but only 50% of the images were oriented and the joining of all the chunks was not good enough). It can be stated that this technology is not suitable for practice, the results are not certain, and the processing is difficult, although it can certainly be used as an experimental procedure.

### 3.2. Laser Scanning

#### 3.2.1. Method Introduction

Laser scanning is a modern technology that many people prefer; laser scanning automatically generates 3D data from the measured area (point cloud), from which the further desired parameters of the imaged objects can be obtained. The classical TLS method uses “scan and go” measurements, i.e., measuring at a position and moving to the next one. The problem generally occurs with joining the data from individual positions into a single point cloud. In recent years, much has been automated based on correlation [20,24].

Many works and articles have referred to the use of TLS in forestry [20,21,22,23,24,25,26,27]. The first task is to set up an optimal scanning distance for the recognition of stems based on the forest type and terrain configuration. According to [23], the most suitable distance between scanner positions is 40 m; however, this research was conducted for beech-dominated forests, where more than 90% of the trees with a DBH of ≥10 cm were successfully detected. 

In the following experiments described below, we used two laser scanners with different parameters

#### 3.2.2. Data Acquisition

The same test area was used for the photogrammetrical experiments. The used laser scanner was the precise Surphaser 25HSX (Basis Software Inc., Redmond, WA, USA), with a point density setting of 5 mm on 10 m with the single-scan approach; the precision of this scanner reaches 0.6 mm of measured distances on 10 m. In the first attempt, four scans were taken, based on the recommendation in [23], in the span of approximately one hour (Figure 5a). The joining of four-point clouds from the marked area of 35 × 30 m was not successful due to the high percentage of hidden areas; the measurement was repeated with nine regularly distributed positions (over more than two hours)—the joining of the point clouds was again unsuccessful. Hidden places caused by other trees were the fundamental problem, despite three of the signalized control points being measured by the total station Trimble 5000 Series. It might have been possible to merge the scans, but after several hours of work, we dropped this because of the labor and inefficiency involved. Of course, some problems could be the type of laser scanner and the parameter setting. Procedures for automatically joining the recently taken scans using spheres (e.g., using the Faro scanner) (Faro, Lake Mary, FL, USA) or correlation (e.g., the Leica BLK360 scanner) (Leica, Heerbrugg, Switzerland) seem to be better. Scanning with a larger number of scans in this small area was completely uneconomical. Other problems were the weight of the scanner; the need for a classic tripod, laptop, and battery; or the regularly distributed and measured control points.

After the unsuccessful first attempt, new measurements were prepared with the new modern Leica BLK 360 laser scanner (Figure 5b), which joins all scans automatically. The immense advantages of the Leica BLK 360 are its low weight and very easy operation. Nine scans from the same study area were obtained (Figure 6). The scanning time was approximately one hour, but the processing time only lasted six hours. Unfortunately, the automatic point clouds joining using correlation did not work correctly because of the object type. It is true that the weather was not ideal and there was a dusting of snow, however, logging often takes place in the winter months. The scan joining was performed manually, resulting in large errors. Hidden areas were still problematic, which limits the usage of laser scanning in forested areas; it causes a low overlapping percentage of scans. Furthermore, there would have to be further processing for stem detection and diameter measurements. For comparison, the same site was manually completely measured over two hours with direct results. 

The approximate DBH could be determined in individual scans, but not in the ground plan (Figure 7). There was a permanent problem here. The hidden parts were unseen by the laser scanning and the structure of the surface (although it contained relatively low vegetation). From the processing report (Table 2), it is clear that, despite the small area covered by nine scans, the overlap was not sufficient (Figure 8).

In 2020, a new measurement with the BLK360 laser scanner was carried out with better weather conditions and more scanner positions (Figure 9). Together, 16 laser scanner positions were used for this small area over two hours. All scans looked very good, but again, the scanner failed in automatic merging of individual scans into a single point cloud. Only three subsets were joined automatically. After several hours of manual processing (data joining), we left this experiment incomplete. The parameters of the TLS experiments are shown in Table 3.

### 3.3. RPAS (Drone)

One often cited modern technology is RPAS. The device is called a drone or UAV (unmanned aerial vehicle), and is more popular as a low-cost device for mapping and monitoring on a local level [15,17]. Of course, from a flight, it is not possible to measure the DBH, but this technology can be used for a CHM (canopy height model), for example. In this case, RPAS is an aerial photogrammetrical technique that uses low-cost cameras for creating an orthophoto and DSM [13]. A bigger RPAS can be equipped with more complex and expensive instruments such as laser scanners or hyperspectral scanners, but the equipment today is still more scientific because of its cost [6,10,23]. As a photogrammetrical result, high-resolution mapping products can be obtained over compact and possibly less accessible areas (e.g., moors, swamps, dumps, and dangerous areas)—or over forested areas [13,16]. At the Czech Technical University, Faculty of Civil Engineering, Laboratory of photogrammetry, we are well equipped with winged drones (eBee and other types) and quadrocopters, which are the most popular and are relatively cheap today (DJI Phantom or DJI Mavic). For large areas (hundreds of hectares), an eBee drone with changeable sensors (RGB, NIR, multispectral Multispec 4C, or a Sequoia camera and thermal camera) can typically be used. Today’s multicopters are typically equipped with a good RGB camera with 4k-image quality. RPASs were used experimentally in our long-term projects to monitor biodiversity, logging and post-logging cleaning, and forest health condition mapping.

In this project, which was focused on deriving forest parameters, both the eBee winged drone, which was equipped with an RGB camera, and three types of quadrocopters (eBee (eBee, Geneve, Switzerland), DJI Phantom 3 and 4, and Mavic Pro (DJI, Shenzhen, China)) were used (see Figure 10). Our project area was ideal for multicopters, but not very good for the fixed-wing eBee (the area was too small). Primarily, it was necessary to obtain a photoset. On the tested area, all possible RPASs were used. As it turned out, during processing, the limitation for a quality DSM and tree detection was the wind. With a wind speed of up to 2 m/s, the photo flight can be conducted, but stronger winds or wind gusts cause the trees to move, and the DSM either fails to generate or is of a bad quality.

#### 3.3.1. RPAS Data Capturing

A photoset captured with an RPAS makes it possible to create a high-resolution orthophoto and DSM, from which it is possible to detect individual trees or, better yet, their number. From our case project, a coniferous forest is better for this purpose. Deciduous trees are usually harder to find due to the involvement of tree crowns. In the Czech Republic, spruce forests are typical and frequent. In the first test flight, even with light winds, the peaks of the trees were swaying and the image correlation between the individual images did not work successfully in this case. The resulting DSM and orthophoto were heavily deformed or contained gaps in their data. For this reason, the measurement had to be repeated during calm weather one or two weeks later with three types of RPAS (Table 4). By using an eBee drone, two flights were achieved (Figure 11). A detailed DSM was calculated from both photosets. After analyzing the derived DSM and orthophoto, it can be said that this output was not good enough for tree detection—the quality of the created DSM was not sufficient in detail (Figure 12). However, the IR orthophoto can be used for other purposes (forest health status and dead tree detection). 

For this small tested area, multi-copters seem to be better to use. DJI Phantom 3 (Figure 13) and 4 were successfully used to collect the necessary photosets. The first flight day was very windy, so the data from DJI Phantom 3 had an insufficient quality for precise DSM creation due to strong winds and the trees moving. Later, the selected area was overflown by DJI Mavic Pro (Figure 14), and after two weeks, the next flights were taken with DJI Phantom 4 at different flight levels (Figure 15). Three flight levels were selected experimentally for the creation of the most suitable outputs. After analyzing the results from all flight levels, it was found that the best results were achieved at the flight level of 55 m (i.e., about twice the height of the trees). The used RPAS and flight parameters are in Table 4.

#### 3.3.2. RPAS Data Processing

This section is focused on determining the number of trees, their heights, and the cubic volume of wood. For this purpose, the data from the RPAS and specifically generated DSM were used. The tree number was calculated based on the local maxima in the DSM using ArcGIS software. The elevation diagram is shown in Figure 16a.

The processing scheme (Figure 16a):Photoset;Agisoft PhotoScan, Pix4D—creation of the DSM and orthophoto;Definition of tree height (subtracting the DRM from the DSM);ArcGIS—analysis of and search for tree position;Input of average tree diameter: the diameter at breast height (DBH);From approximate tree height, typical tree diameter and tree number wooden volume can be calculated.

Agisoft Photoscan and Pix4D version 4.8.0 were selected as the photo processing software. The Agisoft software was used to process the photos from the multicopters (Figure 16b), and Pix4D for the data captured by the eBee. A DSM, DRM, and orthophoto were created in both programs for all flight levels and for a further comparison of the data (Figure 17a–c). Additional computation and analyses were performed in the ArcGIS software version 10.8.2.

Typically, there is a problem with the diameter configuration for finding the local maximum, since it should be obtained experimentally; in our case, it was set to 30 [10]. After this, the selected local maxima were converted from the raster into the point layer, as is shown on (Figure 18a,b). In the selected area, the number of trees was counted. To find the height of every tree, subtracting the DRM from the DSM near the forest–meadow edge was used. When the DBH was added to the previous result, we obtained all the necessary information for calculating the wooden volume (cubature) of that area (Figure 18a,b and Figure 19a,b).

The CHM was used as a basic model for computing the tree height. Basically, the CHM model is the difference between the DSM and DRM. It was calculated and analyzed using the ArcGis software to obtain a local maximum for every tree, the number of trees, and their positions (Figure 20). For the selected research area, an orthophoto was calculated and georeferenced using the eighth GCP, and stabilized by 60 × 60 cm flat signals. They were measured using the precise Topcon FC-100 GNSS device (Topcon, Tokyo, Japan) (cm’s level) and the Trimble 5000 Series total station (Trimble, Westminster, CO, USA) (mainly points in the forest and on the growth edge—there was a poor-quality GNSS signal). The accuracy of the GCPs was approximately 1 cm in position and 2 cm in the vertical direction. All the GCPs were stabilized using a 15 cm long nail in the middle of the signal. For comparison, the height of three typical trees was measured directly by the total station from the observation point. All the measured GCPs were used for the orthophotos’ georeferencing (the typical precision on the GCPs was 1–1.5 multiple of the used GSD, based on the flight height). This was necessary for detecting the same area using terrain measurements.

To verify the aerial measurement, we performed a terrestrial measurement with a common total station and compared them with each other (see Table 5 and Table 6). For the best results, the calculations for every flight level were compared. In the following tables, a comparison of all the measurements is shown. After the analysis, a conclusion could be reached that a 55 m flight height was the best. This is approximately twice the forest height (Table 5 and Table 6).

The terrestrial measurement gave us the average height of the trees, the average thickness of the trees, the number of trees, and, after computing, the cubature of the forest. In Table 6, a comparison of the results from different flight levels with a terrestrial measurement is introduced.

To evaluate the resulting values, the mean absolute error (MAE) was computed by comparing the expected (ӯ_j_) and estimated (y_j_) values. The predicted values were taken from the calculated ground heights, and the values of the calculated heights from the RPAS data were taken as the estimated values. The MAE indicated how the estimated values were close to the predicted values [10]. All the calculations were performed in Excel using the following formula, where *n* is the number of individual trees (Equations (1)–(3), Table 7 and Table 8).
(1)$$MAE=\frac{1}{n}\sum_{j=1}^{n}\left|y_{j}-{\bar{y}}_{j}\right|$$

The accuracy of both measurements was further calculated using the (complete) mean square error RMSE:(2)$$RMSE=\sqrt{\frac{1}{n}\sum_{j=1}^{n}{\left(y_{j}-{\bar{y}}_{j}\right)}^{2}}$$

The RMSE% was calculated as a percentage of the error in the given conditions:(3)$$RMSE\%=100\%\cdot \frac{\sqrt{\frac{1}{n}\sum_{j=1}^{n}{\left(y_{j}-{\bar{y}}_{j}\right)}^{2}}}{\bar{y}}$$

There is another progressive technology that can be successfully used in a forested area—laser scanning on moving equipment, such as MLS (terrestrial) or ALS.

### 3.4. MLS/PLS

MLS technology uses a scanning device on a moving carrier; it can be a car, an ATV, a boat, a train, or a person (a scanner in a backpack or a handheld scanner) [34,35,36,37,38,39,40]. Mobile laser scanning technology for personal use, e.g., a Leica Pegasus backpack or ViaMetris backpack, GreenValley, etc. was introduced some years ago; unfortunately, it is very expensive. A better solution is a handheld scanner—a personal laser scanner (PLS). Nowadays, there is more suitable equipment, such as ZEB-REVO (GeoSlam, Nottingham, UK), BLK2go (Leica, Heerbrugg, Switzerland) (introduced in April 2020), or other devices [35,41,42,43,44].

#### 3.4.1. PLS ZEB REVO

ZEB-REVO is a handheld laser scanner (Figure 21a). It consists of a hand device with a rotating laser head and IMU, a cable connected with a battery pack, and a small computer. This scanner is ideal for the easy documentation of a ragged environment, it weighs less than one kilogram, and has a scanning speed of 43,000 points per second, with an accuracy of 1–3 cm; it is designed for data transfer into BIM (floor plans and building measurements) for forestry, cultural heritage, or underground documentation. Typically, it takes one measurement between a few minutes and 50 min. The ZEB-REVO scanner uses 3D SLAM (simultaneous localization and mapping) technology and works fully automatically. After post-processing in GeoSLAM Hub software version 4.0.1, a 3D point cloud is generated (Figure 21b, Figure 22 and Figure 23), and after this, in Draw module vectorizing, and plans can be generated).

Based on Table 9, the PLS ZEB-REVO instrument is ideal for this type of measurement in forestry; it is easy to use, mobile, light, and accurate enough. There is only one remark—the DBH derived from common laser scanning always gives greater values than those from classic measurements. The reason for this is the bark of trees. Rough and irregular bark increases the measured stem diameter. For forestry purposes, bark is not calculated in terms of wood volume.

#### 3.4.2. GreenValley

The capabilities of mobile laser scanners are evolving rapidly. As more devices are added to PLSs, their usability and price increases. The PLS GreenValley was tested, which is equipped with two wide-angle cameras for the coloring of the point cloud, GNSS RTK equipment, a pair of laser scanners, and SLAM technology. The data are therefore referenced during the measurement and colored in the post-processing. The used instrument was a LiBackpack DGC50 (GreenValley, Berkeley, CA, USA) with a measuring range of 100 m, equipped with two laser sensor VLP16 (accuracy ± 1 cm). Based on the surface types, the generated point cloud had a precision better than 5 cm.

During the measurement campaign in 2022, the same forest area was targeted (Figure 24, Table 10). The data were subsequently processed in the software Lidar360 version 5.0 and Limapper version 2.1. From a forestry point of view, there was an excellent possibility of directly detecting individual trees and calculating their DBHs. This is an extraordinary advance for the forest industry. However, the detection of individual trees depended on their position in the forest stand. The edge trees with branches down to the ground were automatically detected incorrectly or not at all; on the contrary, some weak stems or branches were detected and their DBHs were analyzed (Figure 25). Not all possible parameters were tested, as we focused only on DBH.

### 3.5. ALS

Among laser technologies, in addition to TLS, ALS can also be used. This technology deals with a number of works, and it is certainly applicable to larger forested areas [32,33,45]. Generally, as a professional aerial technology, it is expensive and typically used in cartography for digital relief model (DRM or DSM) creation. For its use in forestry, archaeology, or other branches, a higher density of points is required (tens of points per square meter).

Nowadays, cheaper methods can be used; this means RPASs equipped with a lidar sensor or an ultralight airplane with a medium-expensive lidar sensor. For data referencing, ALS technology needs a very precise INS (inertial navigation system; INS = GNSS + IMU).

In this research, a small airplane with a lidar sensor Riegl VUX1-LR (Riegl, Horn, Austria), joined with an Applanix IMU and precise GNSS receiver, was used (Table 11). It failed to specifically capture the same location. A forested area approximately 1km away from the test area was measured as part of another project (Figure 26a,b).

A comparison with the government ALS project for the whole territory of the Czech Republic with a point density of 1–2 points on m^2^ with the locally measured data had a better density and reached 25 points on m^2^. However, the quality and usability of the data in this case were not ideal. The processing was carried out in the PosPac version 7.5 and RiPropcess version 1.9.2 software.

### 3.6. Tablet (Smart-Phone)

One new technology is a lidar device integrated into a mobile phone or tablet. In April 2020, the Apple iPad PRO was released. It is equipped with a very good camera (12Mpix) and LiDAR scanner, which can be used, for example, for AR (augmented reality) for the precise positioning of objects. Depending on its size, the price starts at approximately 1000 USD. The surprise is that this is really a lidar, as shown in Figure 27a,b and Figure 28. However, the Lidar has a small range (up to 5 m), a relatively large point spacing, and the equipment is designed especially for interior models, ideally for AR.

Recently, experiments have been performed with similar devices in a mobile phone [42]. In this project, a tablet was used to create a 3D model of a forest stand in terms of a possible determination of the DBH. Secondly, it was also used to calculate the volume of a smaller deposit of logged wood.

In our research, we tested the best iPad PRO (equipped with a lidar sensor) on the Czech market in April 2020, directly after this release. First, we tested the functionality of the lidar device. It is focused on the creation of 3D models mainly in rooms (Figure 28). The lidar device implemented in the iPad produced a 3D point cloud, and when using the iPad, the measured 3D points were directly joined to a compact model based on IMU and SLAM technology, and you could see what had already been scanned online.

The point spacing was more than 10cm, which seems to be too much for precise DBH measurements in forestry. When modeling a room of about 5 × 5 m, however, it works well. This means a good 3D model can be created only for a small area; for larger units, the model differs fundamentally when returning to the same place. The experiment was repeated on the test area six times with different speeds of movement for up to 40 min, which was the time taken to slowly pass through the whole area. During the measurements, the stems split, and the model deformed considerably. The results were not satisfactory and this method of measuring DBH cannot be used for the time being. The equipment is not designed for this purpose. (Figure 29 and Figure 30).

In contrast, the use of the tablet for an unaffected estimate of the volume of the smaller logged timber deposit was successful. The accuracy of the calculation was good compared to the classic measurement of the logged wood. However, it is necessary to generate a 3D model (it goes automatically) and then send it to a computer via Wi-Fi. The volume can then be calculated using suitable software. The problem is sometimes the ground. On a flat surface, it is simple—the wood is just cut from the surface. On an uneven surface, it is more complicated and laborious.

However, this technology is definitely included in mobile devices and will probably be used more often in the near future, mainly for AR applications or for defining smaller excavation works.

A good example is a low-cost technology for creating a textured point cloud using a smart phone and RTK GNSS device. It is used, for example, in Pix4D’s version of viDOC (Figure 31) or as videogrammetry software 3D Survay. Unfortunately, an RTK GNSS has not worked well in forests so far and, so for easy measurements, this cannot be used yet [46,47,48].

## 4. Results and Discussion

### 4.1. Results of Individual Methods

#### 4.1.1. Close-Range Photogrammetry

Despite the number of published results [7,8], no reasonable result was reached in this experiment, which failed to create a model that would be useful for measuring DBH. The successful creation of the model depended on the type of forest, the canopy and age of the trees, and the structure of the relief. The branches close to the ground were particularly problematic. However, this type of forest is common in the Czech Republic.

The brief partial conclusion: in our case, IBMR photogrammetry was unsuccessful. In our photogrammetric laboratory, we have used photogrammetric methods for documentation since the 1960s. We can responsibly say that, from an economical, technical, and practical point of view, it is not possible to take hundreds or one thousand photos from such a small area with an uncertain outcome—it is better, faster, and more accurate to manually measure the thicknesses (DBH) using a caliper. It is possible that photogrammetry can be used for another type of forest, but it is generally not economical for this use. From this reason, it can be used as a research or experimental technology only.

#### 4.1.2. TLS

Based on our experiments, TLS is still not an economical or convenient method, despite the existing research articles on this topic [20,21,22,23,24,25,26,27]. Its results depend heavily (like using IBMR) on the forest type, its density, the age of the forest, and the terrain configuration.

The automatic joining of scans did not obtain satisfying results. Without precisely distributed and geodetically measured control points, it is hardly possible to join all scans manually. Geodetical measurement control points in a deep forest are not possible with a GNSS device, and classical geodetical measurements are laborious and require experienced staff.

Considering the cost of the laser scanner and the time for both the data acquisition and processing, this technology is not too usable in practice, although many articles have shown good uses of laser scanning in forestry. The problem is the high purchase price (scanner, software, expert for data processing, and computing hardware), and also the weight and transport of the scanner with traditional TLS. New and easy-to-transport TLS such as BLK360 can be partial answer. However, generally, it is about research into the possibility of using modern techniques. Not all must be economical from a scientific point of view, and in the near future, this can be different. At this time, the two technologies mentioned are interesting, but very laborious or expensive in practice. However, they have been used successfully several times in other research projects [7,8,20,21,22,23,24,25,26,27]. For the type of forest used in this project, it turned out that photogrammetry and laser scanning were not usable for DBH measurement in practice. We consulted with the local forest administration.

#### 4.1.3. RPAS

RPASs are a modern and often researched technology for the documentation of 3D objects. They can be used in forestry for many tasks. In our project, this technology was successfully used for the computing of a CHM and calculating the wooden volume. Easy-to-use and low-cost RPASs can be used for small areas. The accuracy of this technology is sufficient, but for practical use, it still needs trained staff and the necessary software, which is not common for forestry.

There are other possibilities on how RPASs can be used in forestry or agriculture [6,13]. Based on cheap, mobile, and immediately deployable RPAS technology, it is nowadays often used for monitoring and inventory sites affected by winds, or for example, bark beetles. The latter needs near infrared or multispectral cameras. With these instruments, forest health can be monitored (Figure 32). From NIR or multispectral image data, the NDVI (Normalized Difference Vegetation Index) is computed, which is used for health status evaluations. However, an NIR or multispectral camera is not installed on common and cheap RPASs. Using an IR camera, only dead or almost dead trees can be detected. For the searching of trees freshly infested with bark beetles, IR cameras are insufficient, so it is necessary to use expensive hyperspectral cameras or scanners.

#### 4.1.4. MLS/PLS

Based on Table 8, the PLS ZEB-REVO instrument is ideal for this type of measurement in forestry; it is easy to use, mobile, light, and accurate enough. Similarly, LiBackpack DGC50 was tested; it is significantly more expensive, but the software used can directly analyze trees and DBH from the data. However, the automatic analysis failed, especially for the forest edges (Table 9). There is only one remark—the DBH derived from common laser scanning always gives greater values than those from classic measurements. The reason for this is the bark of trees. Rough and irregular bark increases the measured stem diameter. For forestry purposes, bark is not calculated in terms of wood volume. MLS has can be employed for other uses, for example, the easy and precise measuring of wood volume on a wood depot.

#### 4.1.5. ALS

ALS is useful for large areas for wood volume computing. Like RPAS technology, it produces a point cloud, which can be processed based on point density, globally for wood mass measurements, or locally for tree detection. In our project, the point density reached dozens of points per m^2^. After the data processing, the point cloud represents a good approximation of the forested area, which can be used for the wood mass estimation. However, for the presentation and detection of individual trees, it is not very good.

#### 4.1.6. Tablet

The experiment showed that the creation of a quality 3D model for forestry or documentation purposes with this tablet was not possible. An area of 30 × 35 m is too big for this technology; at the beginning, the gradually created model may seem good, but after a few meters of 3D point binding, the technology accumulates errors, and a complex model does not emerge. The result was not sufficient for the measurement of DBH or for precise stems modelling.

### 4.2. Measurement Technology in Forestry—A Short Conclusion

In the following Table 12a, a comparison of the used technologies is shown. The same forested area was measured with different technologies (only with ALS technology, a wider area was overflowed, to approximately 1 kilometer from our test area). In forestry practice, classic equipment still remains and will certainly continue to be used in the near future. Modern methods are usually significantly more expensive and require skilled operators, software, and hardware. In terms of efficiency, speed, accuracy, and low training requirements, PLS can be recommended. In particular, the handheld, lightweight, easily portable, and very simple ZEB-REVO is ideal for forestry purposes. Small and relatively inexpensive drones can also be used to survey and model smaller forest areas. Their use is simple, and many people use them as a hobby. Other methods are problematic; they do not always give good-quality results, need expensive equipment, and require a highly experienced operator. Even relatively cheap and simple close-range photogrammetry does not usually give usable results in forestry.

## 5. Conclusions

This article is critical for the practical application of some technologies (IBMR and TLS), which does not mean that they are not useful. Unfortunately, these technologies did not work on the used typical sample of a spruce forest in the Czech territory, even after many experiments. The Laboratory of Photogrammetry at CTU FCE has been dealing with 3D documentation using photogrammetry and TLS for a very long time and has certainly had experience with them, recently having acquired other modern equipment. From our experiments and based on consultations with typical forestry workers and technicians, it is clear that, for now, although many geomatics specialists will possibly not agree, classical laser scanning (TLS) and close-range photogrammetry (IBMR) in forestry are just technological options. TLS can be accurate, but it is slow and expensive, the results are unclear in advance, and furthermore, trained operators with experience in these technologies are needed. We do not want to say that it does not make sense; it is a research and technology in development. Several similar experiments in other regions have been successful, but in our opinion, they were very laborious and unusable, not only in Czech practice.

The above example study shows that the use of RPASs in forestry is possible, but they are a research and technological possibility too. The counting of the number of trees, their height, and wood volume can be derived from an aerial photogrammetric data set collected by a typical multicopter, such as the DJI Phantom 4, but only in the case of a coniferous forest and in windless weather. The precision of the results depends on the flight height and precision of the DSM and DRM and on the flight level, which determine the GSD. This study shows that the most suitable flight height is about twice the height of the trees under the parameters of the DJI Phantom 4 camera. At a maximum wind speed of 2–3 m/s during flight, with a maximal wind impact of about 5 m/s, it was possible to collect data with a sufficient quality over the forest area. With a typical tree height of 25 m, the data from a flight height of 55 m showed the best quality of all the flights and had the greatest potential for further processing in ArcGIS.

In the Czech Republic, somewhat uncommon methods in forestry, such as ALS and tablet use, were tested. ALS can be a good tool for estimating the volume of timber in large forest areas, but this is not the case in the Czech Republic, where there are often small forest plots and many owners. On the contrary, tablets with lidar cannot be used yet. However, it was verified that the lidar tablet, RPASs, and TLS can be used to calculate the volume of wood on a small stem deposit. Unlike all technologies, mobile laser scanning (such as ZEB-REVO or GreenValley LiBackpackDGC50 mobile hand-operated scanners) excels; ZEB REVO can be used for smaller objects and in forests because it is simple, sufficiently accurate, light, and mobile, and it can achieve excellent results—even after short training—for common forestry technicians. They can also be used to determine the volume of logged wood, and they do not require extremely powerful computer technology or processing knowledge, as is the case with photogrammetric, TLS, or ALS data.

In conclusion, we can state that (especially in the Czech Republic) photogrammetry, TLS, or ALS cannot be used in routine practice, which was found by our own research and especially after consultations with foresters. These methods still remain in the field of research into the applicability of modern technologies.

## Figures and Tables

**Figure 1 sensors-23-07376-f001:**
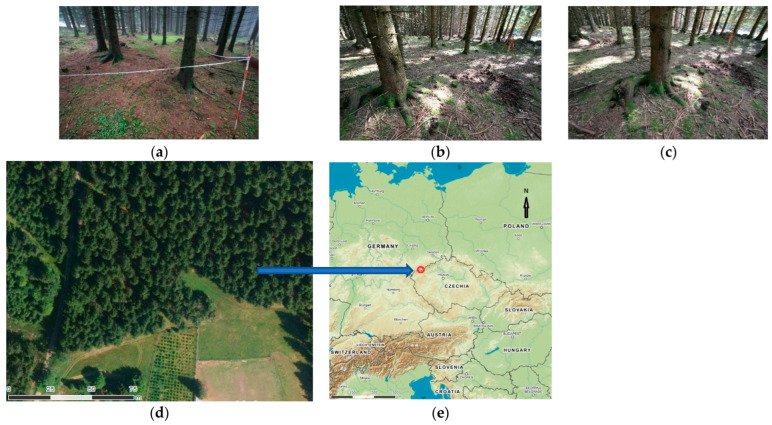
(**a**–**c**) Examples of photographs inside the study area, (**d**) detail of the study area (Ebee drone photo), and (**e**) localization of the study area (GoogleMaps).

**Figure 2 sensors-23-07376-f002:**
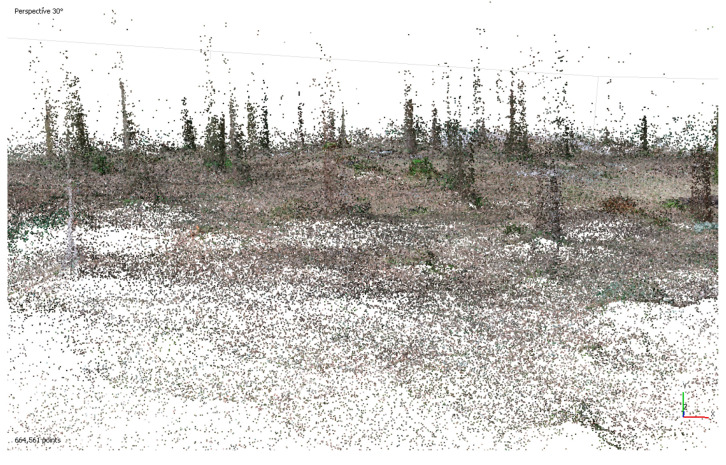
Tie points (sparse point cloud; 665,000 points).

**Figure 3 sensors-23-07376-f003:**
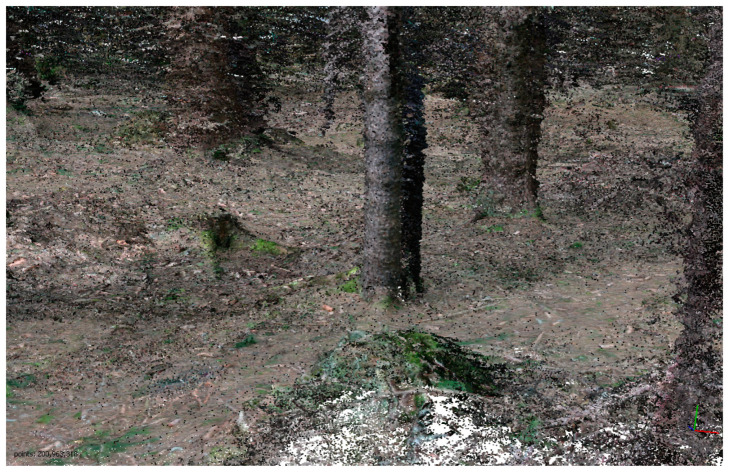
A fragment of dense point cloud.

**Figure 4 sensors-23-07376-f004:**
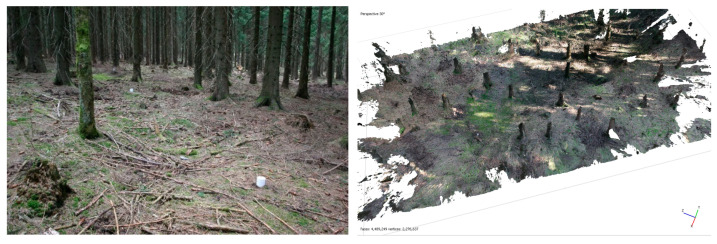
An example from a photo set with control points (**left**), processed model to textured mesh (**right**).

**Figure 5 sensors-23-07376-f005:**
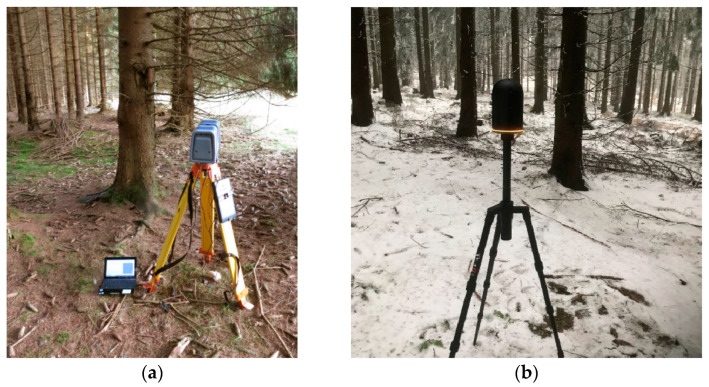
TLS in the tested forest area (**a**) Surphaser 25X and (**b**) BLK 360.

**Figure 6 sensors-23-07376-f006:**
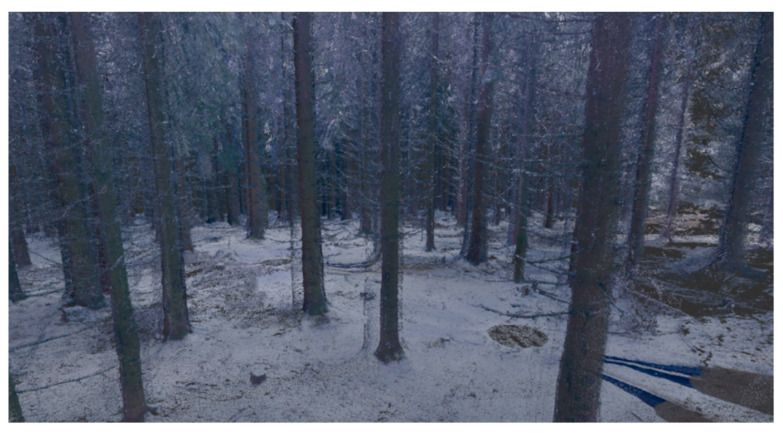
Colored joined point cloud from the laser scanner BLK 360 (520 million of points, 9 scans).

**Figure 7 sensors-23-07376-f007:**
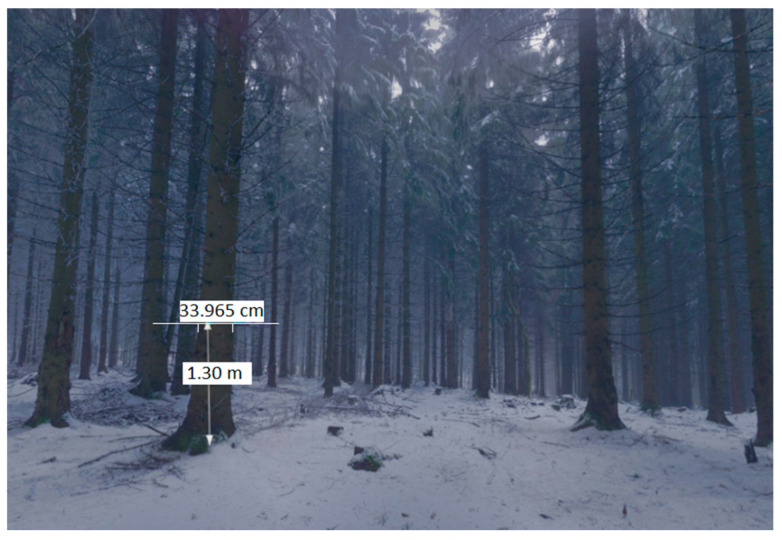
Panoramic model viewer, the possibility of a stem diameter measurement—DBH (BLK 360, Recap software, version 6.2.0.66).

**Figure 8 sensors-23-07376-f008:**
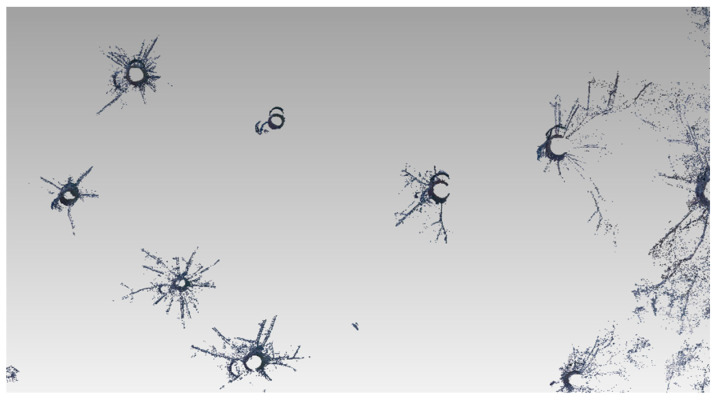
Problems of scan joining—some stems are not joined precisely (there are errors in decimeters).

**Figure 9 sensors-23-07376-f009:**
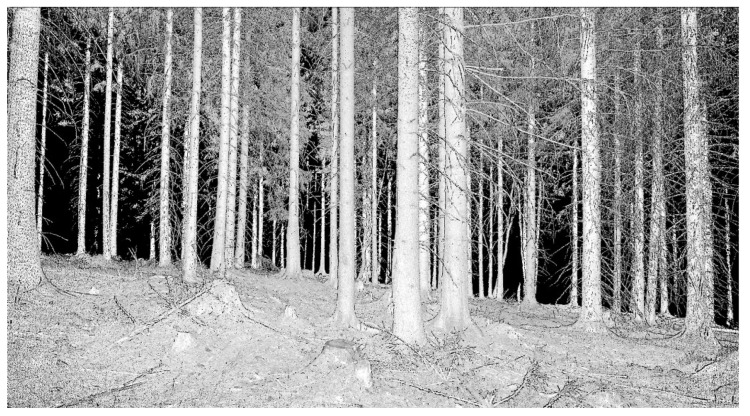
New measurement with BLK360, April 2020.

**Figure 10 sensors-23-07376-f010:**
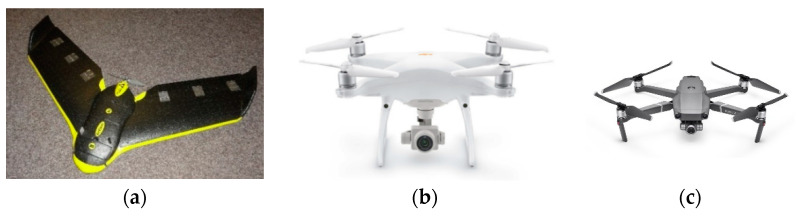
EBee, Sensefly (**a**) DJI Phantom 4 (**b**), and DJI Mavic Pro (**c**).

**Figure 11 sensors-23-07376-f011:**
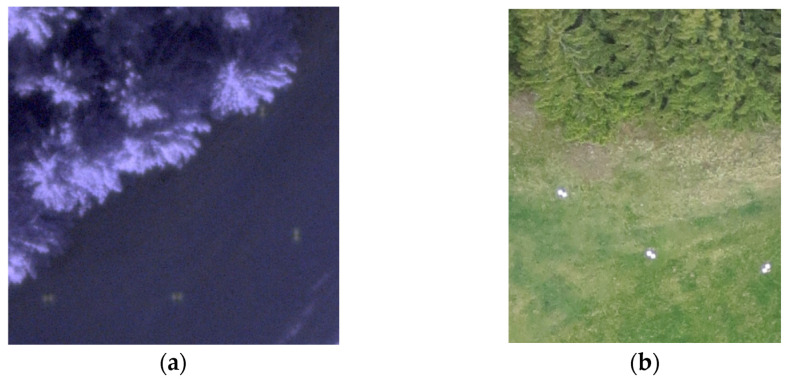
(**a**) A fragment from IR photo (6 May 2018) and (**b**) RGB photo (12 May 2018); both show that the quality of images is not excellent (especially by IR (infra-red) photo, taken in the late afternoon, there are the GCPs (ground control point) poorly visible); eBee cameras Canon S110 IR and ELPH (RGB).

**Figure 12 sensors-23-07376-f012:**
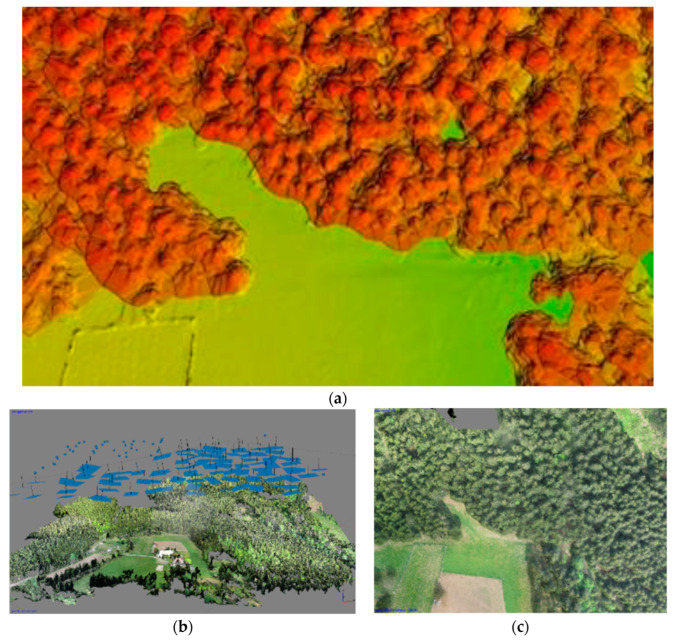
(**a**) DSM from the EBee image set, (**b**) 3D textured model (some images are not processed due strong winds and bad texture), and (**c**) orthophoto.

**Figure 13 sensors-23-07376-f013:**
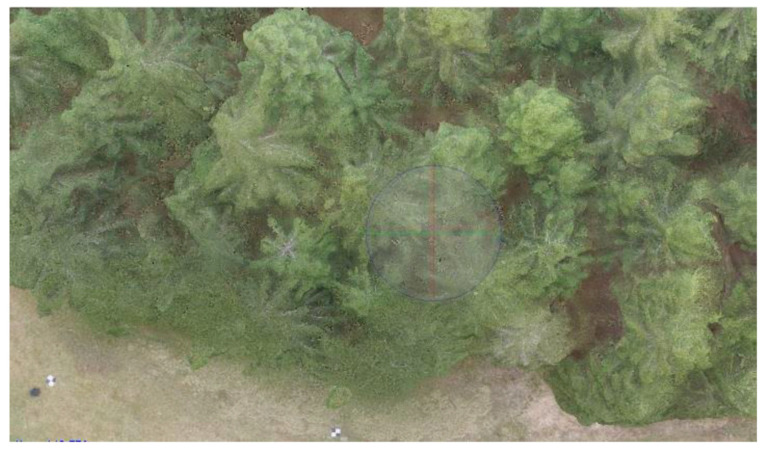
The 3D textured model (derived from a photo set from DJI Phantom 3 on 27 April 2018), a visibly poor quality of the model due to strong winds.

**Figure 14 sensors-23-07376-f014:**
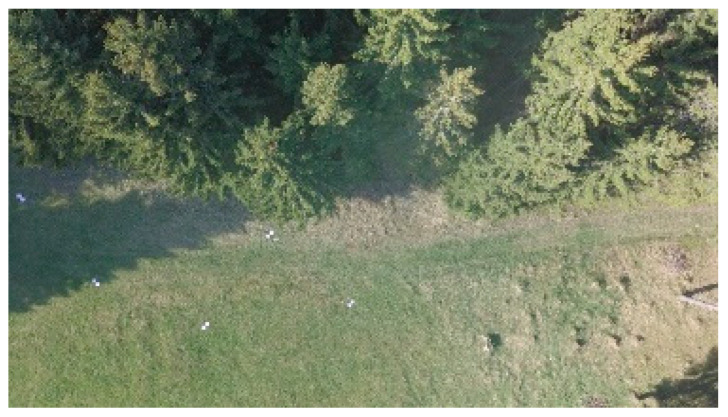
Photo from DJI Mavic Pro, 6 May 2018, (flight level 25 m, GSD = 1 cm—it seems to be to detailed for the creation of DSM based on image correlation).

**Figure 15 sensors-23-07376-f015:**
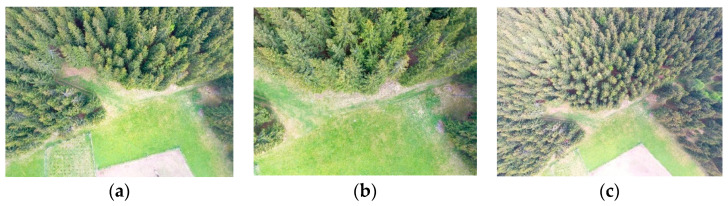
Photos from DJI Phantom 4, 12 May 2018 (three flight levels—(**a**) 30 m, (**b**) 55 m, and (**c**) 80 m).

**Figure 16 sensors-23-07376-f016:**
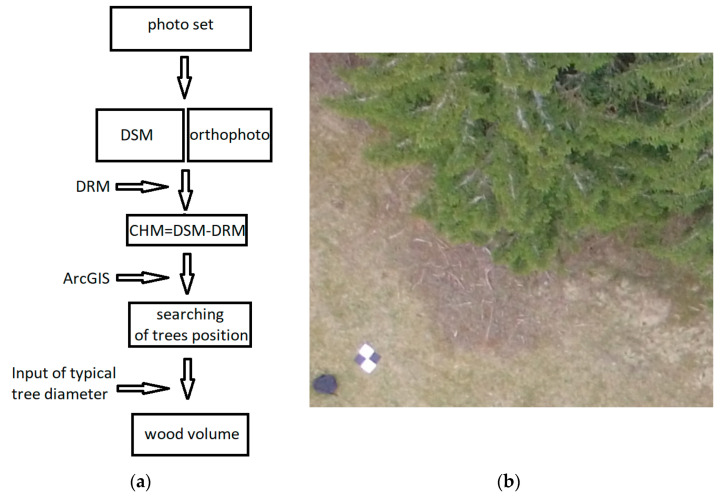
(**a**) Scheme of processing, and (**b**) a fragment of a photo from DJI Phantom 4 with GCP.

**Figure 17 sensors-23-07376-f017:**
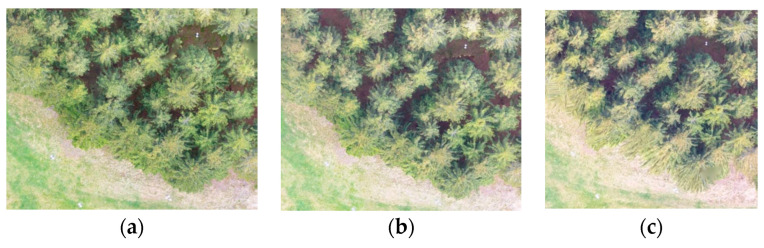
Orthophotos, DJI Phantom 4 from flight altitude (**a**) 30 m, (**b**) 55 m, and (**c**) 80 m.

**Figure 18 sensors-23-07376-f018:**
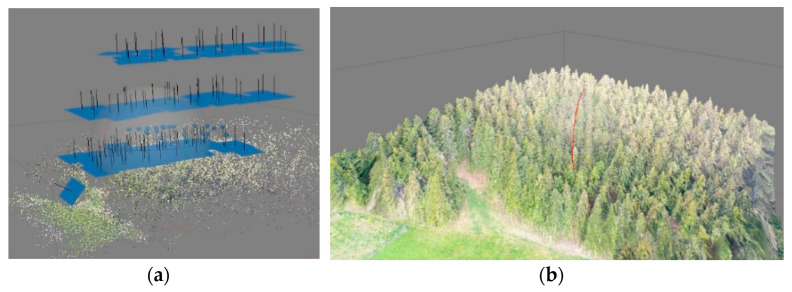
(**a**) three flight levels with DJI Phantom, and (**b**) the 3D model created in Agisoft Photoscan (based on DJI Phantom 4 data).

**Figure 19 sensors-23-07376-f019:**
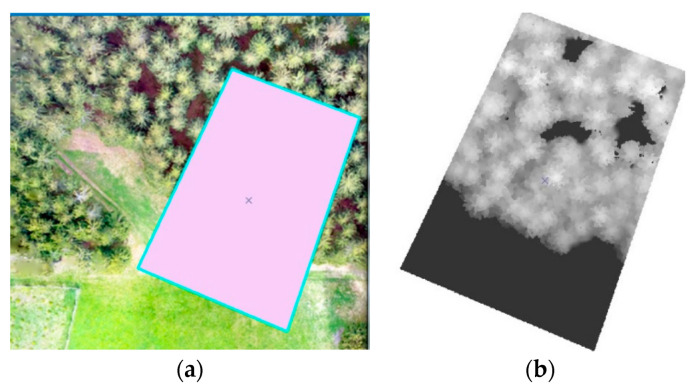
Processing in ArcGIS, (**a**) selected study area, (**b**) CHM (DSM − DRM = CHM).

**Figure 20 sensors-23-07376-f020:**
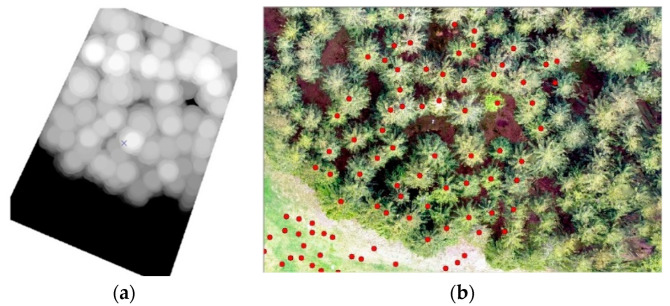
(**a**) Using the function “Focal Statistic” for locating local maxima; in combination with orthophotos, shows the localization of trees. (**b**) Outside of the forested area, errors occurred in local maximum identification (small changes in DSM on meadow).

**Figure 21 sensors-23-07376-f021:**
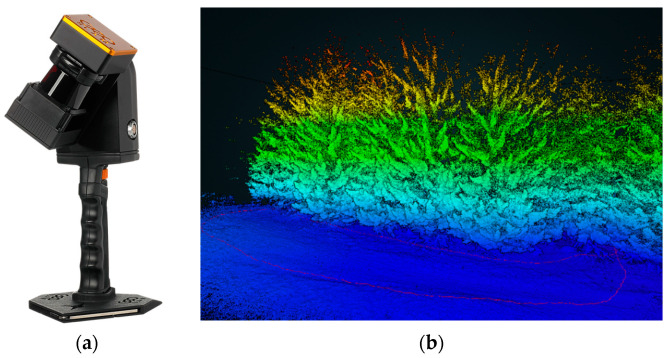
(**a**) ZEB-REVO laser scanner, and (**b**) forest point.

**Figure 22 sensors-23-07376-f022:**
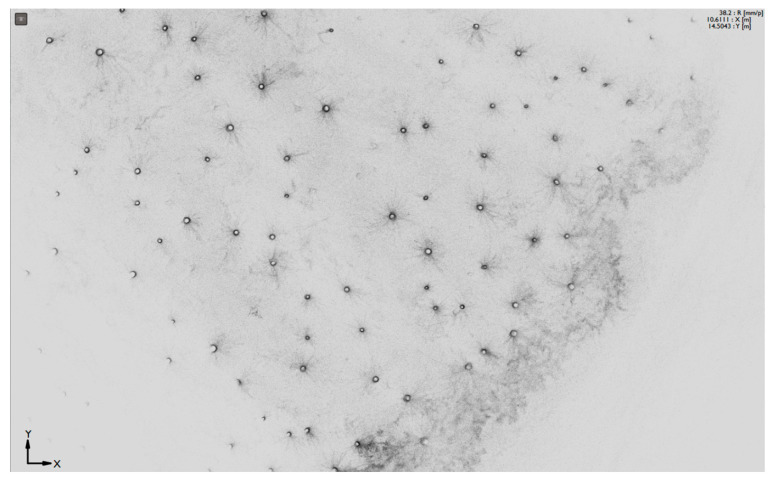
Top view on point cloud from testing area. All stems are easy to find.

**Figure 23 sensors-23-07376-f023:**
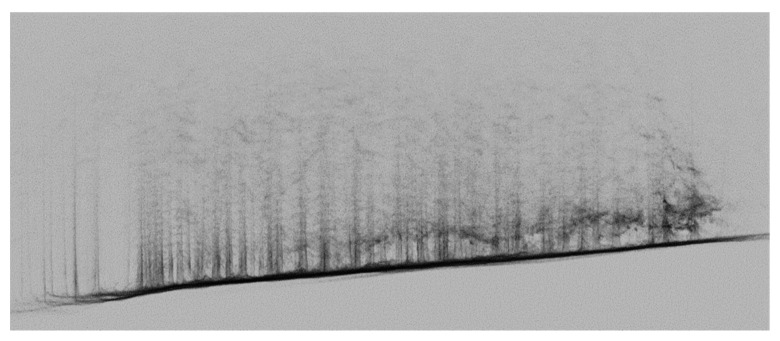
Side view on point cloud. There is possible to make a cut at the height of DBH.

**Figure 24 sensors-23-07376-f024:**
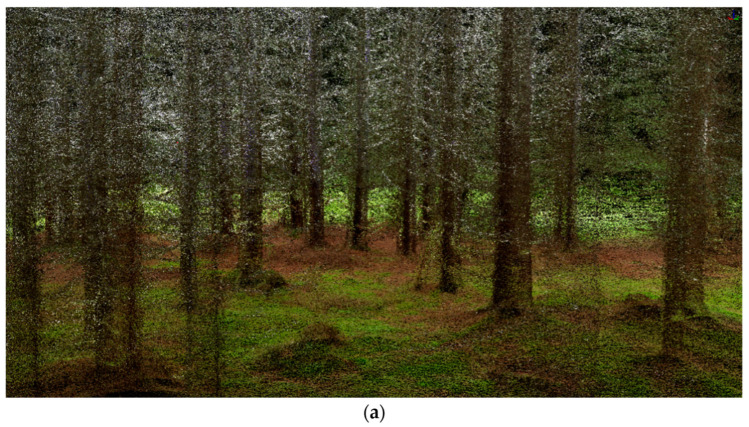
Colored point cloud (**a**) and DSM from LiBackpack DGC50 (**b**).

**Figure 25 sensors-23-07376-f025:**
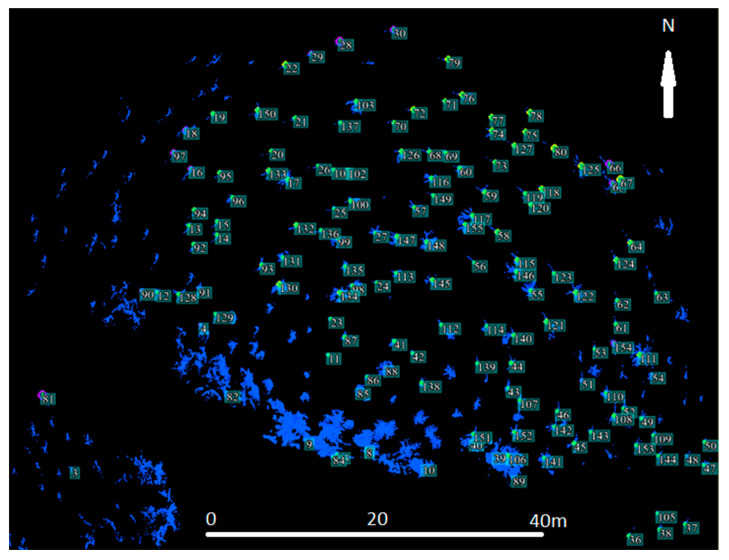
Automatically determined tree stems (Limapper).

**Figure 26 sensors-23-07376-f026:**
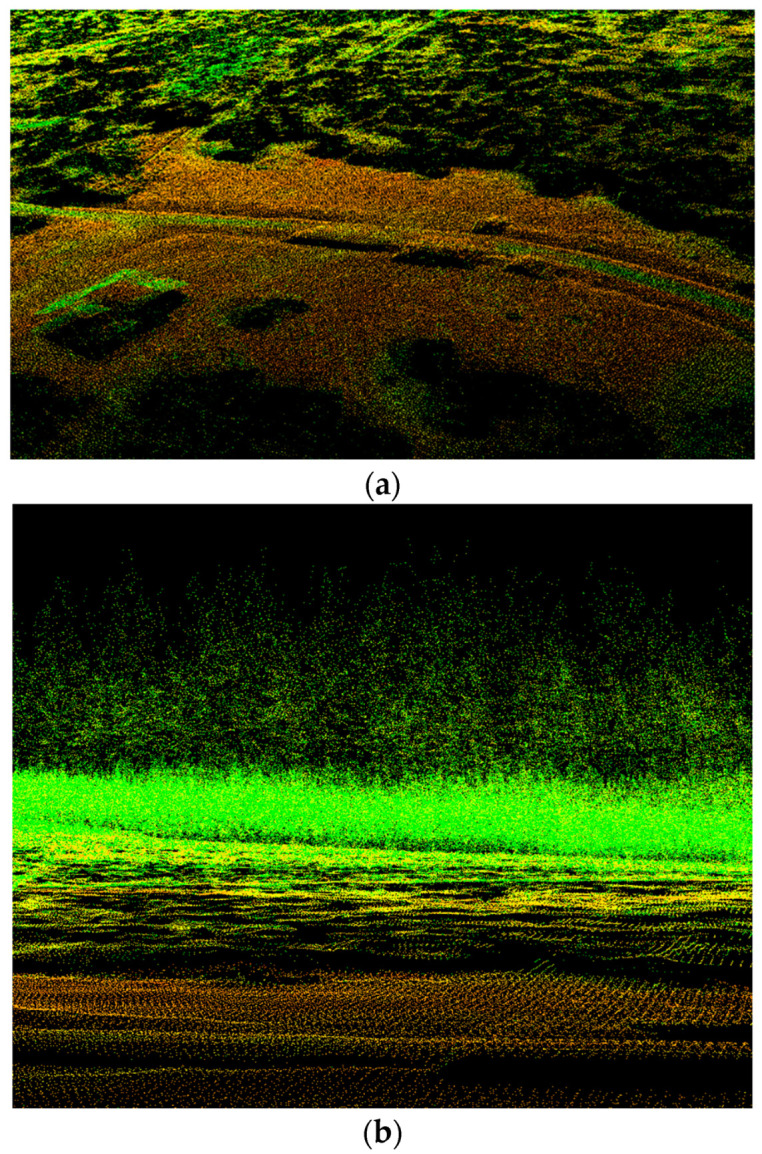
(**a**) A fragment from ALS data set (the oblique view on the test site), and (**b**) ALS data (side view of the test site; you can see the terrain, low undergrowth, and partly individual trees).

**Figure 27 sensors-23-07376-f027:**
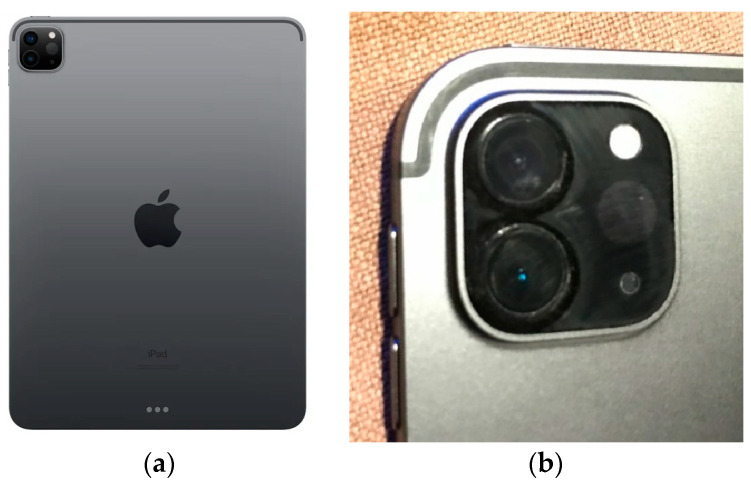
(**a**) iPad PRO, and (**b**) the detail of iPAD PRO lidar sensor and camera lenses.

**Figure 28 sensors-23-07376-f028:**
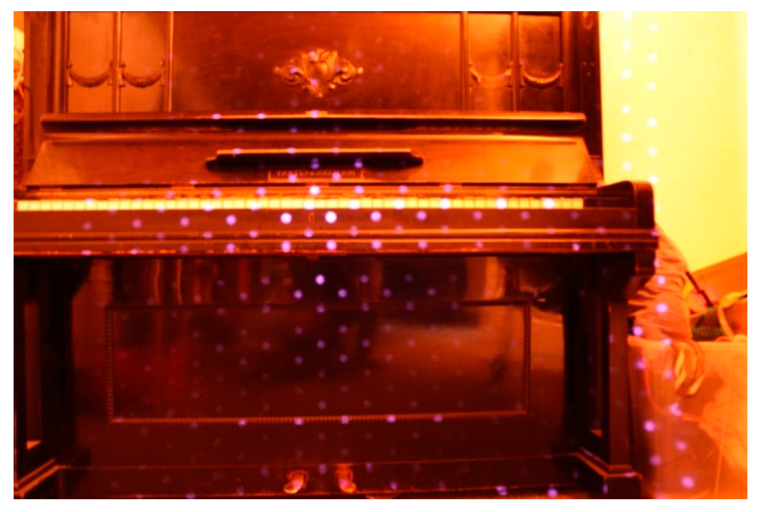
Lidar spots from iPad Pro (infrared image, distance 1.5 m, point spacing approximately 10 cm).

**Figure 29 sensors-23-07376-f029:**
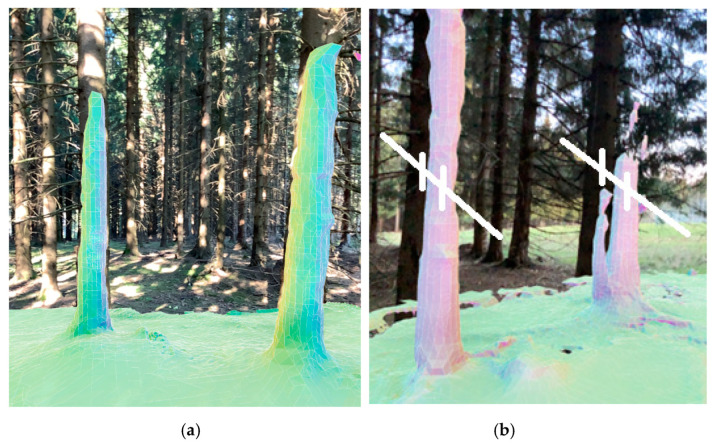
Scanning with iPAD PRO in forest test area: (**a**) left relatively well-scanned stems, and (**b**) after several minutes of scanning, when you return to the beginning, big errors appear (errors reach meters).

**Figure 30 sensors-23-07376-f030:**
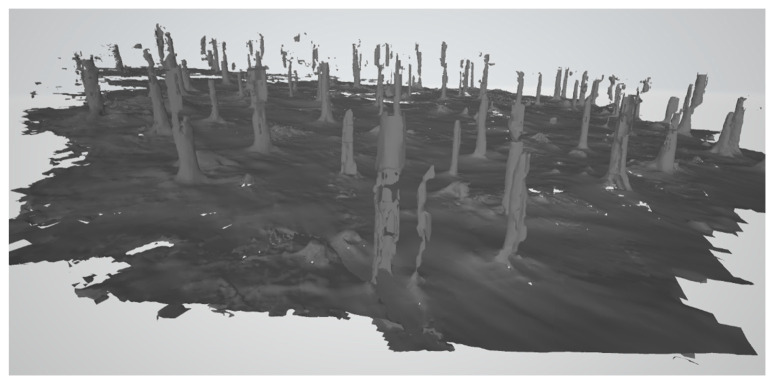
Results from iPhone PRO scanning; a general view on the test site.

**Figure 31 sensors-23-07376-f031:**
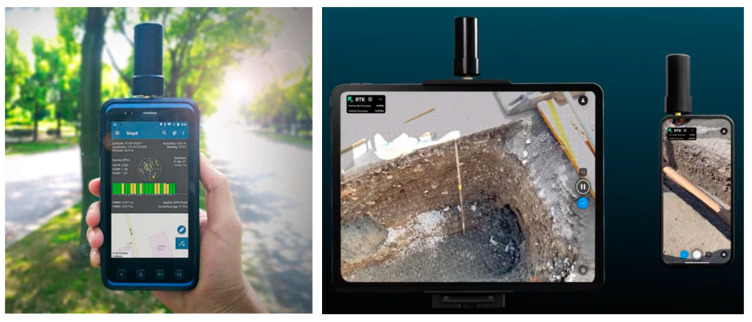
**Left**: Smartphone with RTRK GNSS by using videogrammetry (3D Survay), **right**: viDOC Pix4D iPAD PRO and iPhone PRO.

**Figure 32 sensors-23-07376-f032:**
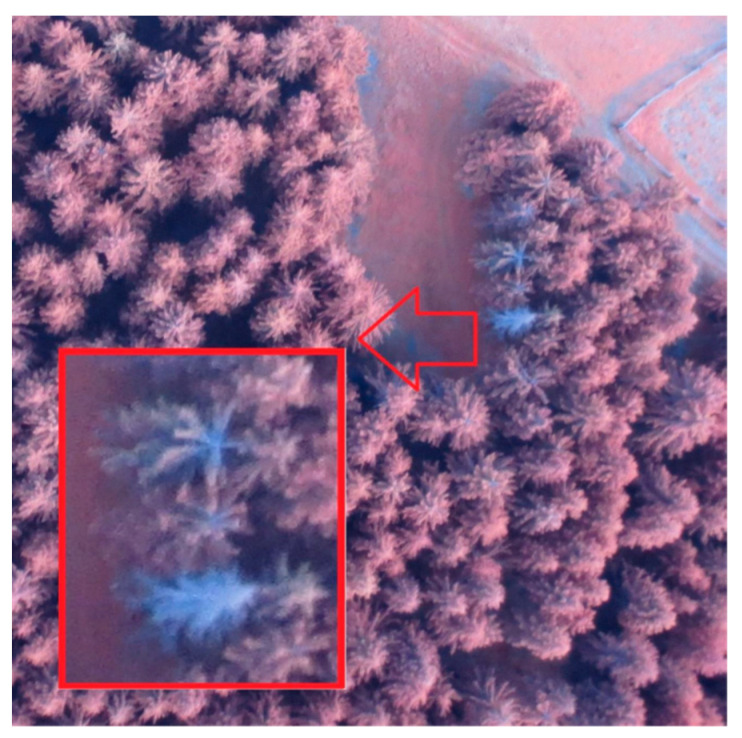
Searching for dead trees (after the attack of bark beetles).

**Table 1 sensors-23-07376-t001:** Parameters of photogrammetric experiments.

Test No.	Date	Camera	No. of Photos	Computer	Photo Taking Time	Photo Processing Time	DBH Measuring/Model
1	24 April 2018	Canon S110/5 mm, 12MPix	280	Notebook i7, 16 GB RAM	30 min	4 h	Not possible/5%
2	25 April 2018	Canon 20D 10 mm	244	HP workstation Z240, 32 GB RAM, i7	30 min	–	Not possible/10%
3	5 April 2019	Canon 20D 10 mm	689	HP workstation	60 min	5 h	Not possible/100% but too big and inaccurate
4	25 April 2020	Canon 20D/22 mm, 8MPix	775	HP workstation	70 min	–	Not possible/18%
5	25 April 2020	Ricoh	515	HP workstation	50 min	–	Not possible/4%
6	28 April 2020	Canon 20D/10 mm, 8MPix	631	HP workstation	50 min	–	Not possible/55%

**Table 2 sensors-23-07376-t002:** Processing report—BLK360, nine scans on forested test area of 30 × 35 m.

Scan No.	Overlap [%]	Balance [%]	Point Accuracy Better than 6 mm [%]
1	7.3	20.5	61.9
2	7.1	13.1	67.1
3	10.3	3.5	82.1
4	2.7	0.5	50.1
5	6.9	8.2	69.0
6	3.8	23.0	67.7
7	5.1	9.2	83.1
8	13.2	5.3	88.5
9	10.0	10.6	88.4

**Table 3 sensors-23-07376-t003:** Parameters of TLS experiments.

Test No.	Date	Instrument	Data Volume [GB]	Precision on 10 m [mm]
1	April 2018	Surphaser 25HSX	4.0	0.6
2	April 2019	Surphaser 25HSX	9.0	0.6
3	April 2019	BLK360	4.5	4
4	April 2020	BLK360	9.6	4

**Table 4 sensors-23-07376-t004:** Used RPAS and flight parameters (GSD—ground sample device).

Flight No.	Date	RPAS	Camera	Flight Altitude	No. of Photos	GSD (cm)	Comment
1	14 April 2018	DJI Phantom 3	12.4MPix	45	130	2	Poor model due to windy weather
2	27 April 2018	eBee	Canon S110 IR, 16MPix	130	106	4	Poor model due to windy weather
3	27 April 2018	eBee	IXUS/ELPH RGB, 16MPix	130	104	4	
4	6 May 2018	DJI Mavic Pro	12.4MPix	40	145	1	
5	12 May 2018	DJI Phantom 4	12.4MPix	30	125	1	
		DJI Phantom 4	12.4MPix	55	98	2	
		DJI Phantom 4	12.4MPix	80	65	3,5	

**Table 5 sensors-23-07376-t005:** Comparison of tree heights from terrestrial and aerial measurement (H_all_ is the processing of all image data together; H_T_ is the terrestrial measurement).

Tree Number	H_30m_ [m]	H_55m_ [m]	H_130m_ [m]	H_ALL_ [m]	H_T_ [m]
1	21.0	21.2	21.5	20.5	21.5
2	22.7	22.6	21.8	20.8	23.3
3	23.8	23.8	22.2	21.7	23.1

**Table 6 sensors-23-07376-t006:** Measured and calculated values to determine the wood volume.

	Terrestrial Method	Flight_55_	Flight_330_
Average height of trees [m]	22.60	23.32	20.97
Average thickness of trees [cm]	29.40	30.40	30.40
Number of trees [number]	76	79	80

**Table 7 sensors-23-07376-t007:** Values for all calculated errors.

Flight Level	MAE [m]	RMSE [m]	RMSE%
Flight_30_	0.60	0.61	2.69
Flight_55_	0.57	0.60	2.65
Flight_130_	0.80	1.01	4.63

**Table 8 sensors-23-07376-t008:** Number of trees in each thickness step.

**Thickness**	14	18	22	26	30	34	38	42
Number of Trees	1	4	12	15	16	16	8	2

**Table 9 sensors-23-07376-t009:** Comparison of classic measurement with a caliper and laser scanning with ZEB-REVO.

	Caliper	ZEB-REVO
Detected trees	76	76 (manually)
Average diameter (DBH)	29.4 cm	30.2 cm
Time of measurement and analyses	2 h measurement and 30 min data analyzing	18 min scanning, 20 min processing, and 20 min DBH manual measurement

**Table 10 sensors-23-07376-t010:** Comparison of classic measurement with a caliper and laser scanning with LiBackpack DGC50.

	Caliper	LiBackpack DGC50
Detected trees	76	67 (automatically)
Average diameter (DBH)	29.4 cm	31.3 cm
Time of measurement and analyses	2 h measurement and 30 min data analyzing	20 min scanning and 20 min processing

**Table 11 sensors-23-07376-t011:** Riegl VUX1-LR parametres (FOV—field of view).

Instrument	Date	Flight Altitude	FOV	Strip Overlapping	Strip Distance	Scan Rate
Riegl VUX1-LR	18 November 2019	300 m	80°	50%	200 m	48 lps

**Table 12 sensors-23-07376-t012:** Comparison of used technologies for DBH or volume of wood measurement.

Technology	Approx. Price per Unit (Euro)	Time for Data Capturing	Time for Data Processing	DBH	Precision	Data Volume	Usability in Practice
Caliper	20	2 h	2 h	yes	cm	1 kb	yes
IBMR	1000+ software	0.5–2 h	Several hours	problematic	mm to cm	5 Gb	no
TLS	40,000-80,000+ software	1–2 h	Several hours	problematic	mm to cm	10 Gb	sometimes
PLS	40,000 (ZEB REVO)	18 min	0.5 h	yes	1–3 cm	35 Mb	yes
PLS	100,000 (LiBackPackDGC50)	20 min	0.5 h	yes	1–2 cm	2 GB	yes
ALS	100,000+ airplane+ software	Several minutes + flight time to area	Several hours	no	5–10 cm	Gbytes generally, a part 250 Mb	partially yes
RPAS	1000 to 30,000+ software	10–15 min	Several hours	no	5 cm	3 Gb	yes
tablet	1500	40 min	Several hours	problematic	5 cm	30 Mb	no

## Data Availability

Data is available after reasonable request, please contact corresponding autor Karel Pavelka (pavelka@fsv.cvut.cz).

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
