# Peer review of "Remarks on Geomatics Measurement Methods Focused on Forestry Inventory"

_sensors, 2023, doi:10.3390/s23177376_

Round 1

Reviewer 1 Report

I found this work really interesting also as an overview of the available information and technology advancements on the topic.

I find it also really useful for environmental implications and i would recommend to have a final look on the available technologies to overcome the little gaps you highlighted and experienced during the field testing.

Author Response

Dear Reviewer, Thank you very much for your opinion.

Reviewer 2 Report

The manuscript is devoted to the analysis of currently existing forest inventory methods and the determination of the method most suitable for small forestry enterprises. The authors tested and intercompared on the data of their own measurements in the Czech Republic several classical and relatively new methods for determining such quantitative forest parameters as wood cubature, diameter at breast height, trees height, and number of trees, and density of growth. The advantages and disadvantages of each method were identified. The main criteria for evaluating each of the methods were their applicability to small forestry enterprises and the price-quality ratio. I believe that the results presented in the manuscript are useful and make a valuable contribution to this research area.

The manuscript material and results fall within the scope of Sensors journal and will be of interest to the remote sensing and forestry communities. My recommendation is to publish the paper in the Journal after some revisions.

General comments

English

English needs to be improved and carefully proofread, for example, by a native speaker (see please the highlighted version of the manuscript below).

Abbreviations

Some abbreviations in the text are not entered properly. In accordance with the requirements of the journal, all abbreviations must be disclosed at their first mention both in the abstract and in the main text. The article has a large number of abbreviations (more than 20), which are sometimes difficult to trace in the text. I propose to create a section in which all used abbreviations would be collected in alphabetical order, for example, after keywords.

Section 2 Project description

I think that Section 2.1 should be included at the end of Introduction (maybe as Section 1.9). I also believe that there is no point in breaking Section 2 into subsections.

Section 2.1 is followed by Section 2.3. Where is Section 2.2?

Section 4 Results

Since there is no "Discussion" section in the article, I propose to call Section 4 "Results and Discussion".

4.2 Measurement technology in forestry – a short conclusion

I believe that the authors should at least briefly discuss the results presented in Table 10.

Figures

Figure 1. All five parts of Figure 1 should be labeled and explained accordingly in the figure caption. For example, what is the difference between the images on the middle and right side of the top of Figure 1?

Figure 7. Please enlarge the font size in the captions in this figure, if possible.

Figure 20 is not mentioned anywhere in the article.

Figure 26 requires a more detailed description.

Figure 30. I did not find parts (a) and (b) in figure 30 and errors in meters.

Technical comments, suggestions, questions, and recommendations

All my comments, suggestions, and recommendations can be found in the highlighted version of the manuscript after the review.

All comments regarding English are given in the highlighted version of the manuscript.

Author Response

Dear Reviewer, Thank you very much for your opinion.

Please, see attached file.

Reviewer 3 Report

The paper provides a comprehensive comparison of modern geomatic technologies used for analyzing growth parameters in forestry management. The authors conducted a critical analysis of the practical application of various technologies for assessing tree growth parameters. The study reveals that drones prove to be an effective tool for estimating timber quantity and conducting health inspections of forests.

The reviewer emphasizes the high-quality presentation of the material, including the provision of qualitative graphical explanations. In the opinion of the reviewer, the article can be published without any revisions.

Author Response

(The authors gave the same response as above.)

Reviewer 4 Report

1. The article cites more existing methods and lacks its own novelty.
2. The summary is too long and introduces too many methods. It is suggested that the authors should simplify the abstract to highlight the novelty of the study.
3. The format of picture captions is not uniform.
4. There is little literature in the past three years, and the latest research should be cited.
5. In section 3.2 Laser scanning, the content is suggested to be divided into two parts: method introduction and data acquisition.
6. None of the methods mentioned in section 3.4MLS/PLS are cited in the literature.

OK

Author Response

(The authors gave the same response as above.)

Round 2

Reviewer 4 Report

accept